# Clinical Feasibility Study of Gold Nanoparticles as Theragnostic Agents for Precision Radiotherapy

**DOI:** 10.3390/biomedicines10051214

**Published:** 2022-05-23

**Authors:** José Antonio López-Valverde, Elisa Jiménez-Ortega, Antonio Leal

**Affiliations:** 1Departamento de Fisiología Médica y Biofísica, Facultad de Medicina, Universidad de Sevilla, 41009 Seville, Spain; jlvalverde@us.es (J.A.L.-V.); elisajimenez@us.es (E.J.-O.); 2Instituto de Biomedicina de Sevilla, IBiS, 41013 Seville, Spain

**Keywords:** AuNP, GNP, theragnosis, precision radiotherapy, dose enhancement

## Abstract

Background: Gold nanoparticles (AuNP) may be useful in precision radiotherapy and disease monitoring as theragnostic agents. In diagnostics, they can be detected by computerized tomography (CT) because of their higher atomic number. AuNP may also improve the treatment results in radiotherapy due to a higher cross-section, locally improving the physically absorbed dose. Methods: Key parameters values involved in the use of AuNP were imposed to be optimal in the clinical scenario. Mass concentration of AuNP as an efficient contrast agent in clinical CT was found and implemented in a Monte Carlo simulation method for dose calculation under different proposed therapeutic beams. The radiosensitization effect was determined in irradiated cells with AuNP. Results: an AuNP concentration was found for a proper contrast level and enhanced therapeutic effect under a beam typically used for image-guided therapy and monitoring. This lower energetic proposed beam showed potential use for treatment monitoring in addition to absorbed dose enhancement and higher radiosensitization at the cellular level. Conclusion: the results obtained show the use of AuNP concentration around 20 mg Au·mL^−1^ as an efficient tool for diagnosis, treatment planning, and monitoring treatment. Simultaneously, the delivered prescription dose provides a higher radiobiological effect on the cancer cell for achieving precision radiotherapy.

## 1. Introduction

The role of radiotherapy (RT) increased its prominence in relation to surgery and chemotherapy in the treatment of localized solid tumors, the most widespread expression of cancer. Among other reasons, this prevalence of RT is due to technological advances in devices associated with delivering and shaping the dose distribution, adapting the beam to the tumor volume, while safeguarding healthy tissues. Imaging devices for RT have also evolved considerably. In RT, the medical image plays a double role: it informs the specialist for a better diagnosis and serves as the basis for therapy by planning the optimal use of radiation sources and calculating the dose distribution in the patient. For this very reason, image-guided radiotherapy (IGRT) uses scans installed in the lineal accelerator to monitor treatment through the sessions, although in most cases the image is only used to verify the correct position of the patient with respect to the beam source. Imaging in RT is usually limited to three-dimensional anatomic information from conventional computerized tomography (CT), and hardly ever from magnetic resonance imaging (MRI). Indeed, including morphological images without considering biological or functional considerations is one of the main current limitations in conventional RT. Therefore, dose prescription and treatment planning are approaches used in evidence-based population medicine, and biological considerations are scarcely included in the planning process using parametrized mathematical models to predict the outcome of RT based on dose distribution characteristics [1,2]. Ling et al. [3] introduced the concept of biological target volume (BTV) as one or more sub-volumes that are usually included in the macroscopic tumor volume identifiable in the image, called gross tumor volume (GTV). The best way to provide precise radiotherapy treatment to improve clinical outcomes is by including biological information about each patient in the image for diagnosis and planning, following a theragnostic methodology [1]. Scientific progress in functional imaging is making RT guided by theragnostic imaging a realistic goal for the next few years, but some aspects have not yet been solved.

Over the last decades, nanoparticles have become increasingly popular in medicine as theragnostic agents for cancer [4]. In RT, inorganic nanoparticles based on high atomic number (Z) arouse a singular interest for clinical application due to their potential to achieve two synergistic goals: improving imaging contrast for diagnosis [5,6,7] and increasing dose efficacy for therapy [8,9]. The basis of using high Z nanoparticles is that they have a cross-section higher than the usual atoms composing tissues, and therefore, when radiation interacts with them and pulls out an inner shell electron, an electrons cascade can be triggered, producing 10–20 low energy electrons, the so-called Auger electrons and Coster–Kronig electrons with paths short enough to ensure the deposition of energy inside those cells that internalize the nanoparticles [9]. On the other hand, the high Z also permits tumor definition using CT imaging, as those images are generated mainly due to the photoelectric interaction, which depends on Z to the third exponent.

Among several nanoplatforms proposed for RT, gold nanoparticles (AuNP) have been the most extensively studied due to their high X-ray absorption coefficient (Z = 79), easy synthesis, and because they are biologically well tolerated and have shown to accumulate passively in tumors while remaining outside healthy tissue when their longest dimension is less than 100 nm [9]. In this way, AuNP represents a theragnostic research area of great interest in oncology [10,11].

Nanoparticles can target tumor cells by two methods: passive targeting, which means the accumulation of specific nanoparticles by EPR effects depending mainly on size and shape, and active targeting, that is, targeting by means of the functionalized surface, using peptides or antibodies that express specific affinity for a type of interaction [12]. Fortunately, X-ray attenuation is not influenced by the size of the nanoparticle, so that parameter would have to be adjusted only to achieve better biodistribution [13]. Furthermore, another reason to choose AuNP as contrast agents is that they exhibit a longer vascular retention time compared to other contrast agents, along with their biocompatibility and AuNP’s molecular weight, resulting in lower toxicity [14,15,16].

As can be seen, the usage of AuNP as contrast agents has been fairly well-studied. As low X-ray energies are used, i.e., around the hundred keV, the main effect of interaction with radiation is the photoelectric effect, with a probability of interaction depending on Z potentially, according to Bragg–Pierce law. Therefore, changes in Z result in a natural contrast in the image caused by the proper attenuation of the X-rays. Thus, gold is an excellent contrast agent for CT and X-ray-based techniques. Unfortunately, some publications ignore the influence of the environment around the AuNP, which can change the beam attenuation, as is the case in the work by Aydogan et al. [17], where the contrast was tested inside a Falcon tube without considering the unavoidable attenuation of the X-rays of the surrounding tissues.

At this point, it seems clear that a coherent methodology should seek to find the minimum mass concentration capable of providing a reasonable contrast level in the CT image with the lowest radiation dose possible, and then try to control the biological parameters such as AuNP biodistribution and tumor uptake through an injectable concentration free of inducing acute and/or long-term toxicity. It was previously thought that some nanoparticles could perpetually remain intact in some tissues. However, Balfourier et al. showed that small AuNP were indeed degraded in vitro by cells through lysosomes, forming aurosome-like structures [18].

Fortunately, AuNP provides many opportunities to find the right formulation with targeted delivery, colloidal stability, and the blood retention necessary for a time-optimal imaging and treatment delivery processes [19]. With respect to these parameters involved in biodistribution and targeting, it seems that some consensus range of values has been determined, such as the size of AuNP, which should be as large as possible to maximize the mass concentration, and smaller than the diameter of the vascular pores (100 nm) for passive or active tumor targeting [20]. On the other hand, larger AuNP showed slower maximum accumulation (8 h after injection for 100 nm nanoparticles) and a gradual decrease after 24 h, as opposed to a continuous increase in accumulation for smaller particles (15 nm) [21], so a size compromise must be established. Polyethylene glycol (PEG) is one of the most widely studied surface functionalizations; it appears to improve colloidal stability, resulting in a longer half-life of the blood 2–4 h, enough to achieve the mass concentration necessary for contrast. These time-lapses are longer compared to other conventional contrast agents, such as iodine, with a half-life of minutes, and could be efficient for a posterior radiation sensitization during treatment with the same AuNP if planning can be done in that time.

Regarding the therapeutic efficacy of nanoparticles, several reviews [8,9,22] collected enough published works showing dose enhancement under different beam modalities and generally expressed the potential efficiency as a percentage of the dose delivered in the same scenario but without nanoparticles. Radiation modality plays an essential role because the results could be substantially different depending on the incident particles—photons, electrons, ions, or protons. Energy is also a critical factor to study. For photon beams, the most clinically extended modality, energy values higher than 1 MeV can provide an adequate depth dose but a low interaction with AuNP in the Compton range. Lower energies provide a higher probability of photoelectric absorption and a higher interaction with gold in cells, but dose enhancement might not compensate for loss of penetration through tissues [8].

Despite this, keV beams have been used recurrently for preclinical studies with relatively small animal models [23,24], and therefore those results were hardly transferable to clinical reality because although the physiological model could be good, the physical considerations are not met. For example, the size of the animal is a determinant for the tumor depth, and hence for the irradiation beam characteristics.

The need for beam interaction modeling implementation in the methodology for studying AuNP as theragnostic agents in a first physical stage appears to be clear. It is essential to accurately describe the energy spectra of therapeutic beams within the patient. Many works have focused on Monte Carlo (MC) simulations [8], which allow for an explicit characterization of the beam modality. MC is already a recurrent tool in RT for the calculation of the dose of complex cases under special RT techniques. Analytical algorithms implemented in treatment planning systems are considered only approximations based on experimental measurements taken previously under standard conditions [25,26]. It is important to note here that although some radiobiological models have been implemented in MC simulations [27], only the physical and physicochemical processes of radiation interactions can be simulated with enough precision, because the biological processes are difficult to describe using a probability distribution function, which is required for the numerical method. Therefore, dose enhancement could be calculated using MC modeling, but it may differ from the radiosensitization effect.

Dose enhancement does not directly predict the biological effect, which depends on additional factors beyond the energy delivered at a specific location. Radiosensitization is dependent on factors other than strong photoelectric absorption, such as the increased Auger emission occurring from de-excitations of irradiated Au atoms. These electrons are strongly involved in the production of reactive oxygen species (ROS), such as hydroxyl radicals, superoxide, and hydrogen peroxide produced from the radiolysis of water, which can cause indirect damage to cellular DNA or may lead to the oxidation of lipids and proteins to generate apoptosis and necrosis of cancer cells [28]. Several in vitro studies showed significant levels of ROS depending on AuNP size, shape, and synthetic surfaces. These results are even time-dependent, so the mechanism involved in AuNP-induced oxidative stress in RT remains unclear [29].

Anyway, to assess the radiosensitization effect, considering chemical-specific ligands becomes an essential factor involved in DNA damage. Unfortunately, these biological processes are difficult to measure in in vitro experiments [30] and cannot be analyzed using in vivo studies, which requires clinical transfer to assess biological parameters such as toxicity or biodistribution. Several works have been published using CT images of mice tumor xenografts injected with AuNP with different properties to be irradiated with different energy beams X-rays [31,32,33].

As commented, despite the high number of publications and the variety of disciplines in which both theoretical studies and in vitro or in vivo experiments are focused, many of these works lack a realistic discussion of the clinical possibilities of AuNP in RT. The variability of the results depending on the parameters evaluated draws attention, from physical AuNP properties to chemical/biological functionalization or beam modality for irradiation. Sometimes, the results of contrasting studies were not relevant because they were carried out using nano-CT instead of the CT resolution applied to humans.

Moreover, previous work based on animal models obviated the benefit of the relative distribution of dose for safeguarding healthy tissue beyond the potentially enhanced dose in the therapeutic target. Kuncic et al. [8] indirectly suggested this when discussing the advisability of showing key parameters such as tumor vs. normal tissue uptake ratio. The relative dose distribution obtained with a treatment planning system in which the presence of nanoparticles can be considered in the functional image of the patient is also a necessary methodology to assess the real possibilities of the nanoparticles in a clinical approach. Anyway, due to the large variety of possibilities for each one of the parameters involved in the process, theoretical modeling can save the time spent on non-necessary experiments and help to understand how these parameters interact with each other. In this sense, our group has previously developed MC-based software [34,35,36], which means having an excellent tool for providing some relevant contribution to the topic.

The clinical implementation of AuNP is a problem that depends on many factors interacting with each other. As far as we know, previous works provided results by using all or some of the aforementioned parameters, albeit with values far away from the clinical application, which disables the capability of actual implementation. It would be desirable to find a set of parameters that could be ready in the clinical routine and capable of improving diagnostic and therapy enough for undertaking a clinical trial.

The objective of this work is to perform a clinical feasibility study of gold nanoparticles as theragnostic agents for radiotherapy treatments and evaluate the radiobiological benefit by following a heuristic methodology. In this way, actual clinical conditions were imposed at the start point to find the optimal values of the key parameters involved in the use of AuNP in RT for diagnostic and therapy. Then, we were able to show the radiobiological key points in in vitro experiments as the previous step before performing in vivo experiments involving the adequate animal model.

## 2. Materials and Methods

For designing a robust methodology, we decided to first find the specific mass concentration of AuNP required at the tumor site to obtain a contrasted CT image. This concentration was named radiological concentration. Considering that not all nanoparticles will reach the cancerous tissue, a dose with a quantity of nanoparticles higher than this radiological concentration will be necessary for patients. Beyond the scope of this work, an in vivo experimental stage should evaluate pharmacodynamic and toxicity aspects to impose a clinical concentration, so our approach consisted of handling different concentrations until the lowest relative contrast level was achieved.

Nevertheless, a higher concentration may be necessary to achieve the expected enhanced therapeutic effect. This concentration, we called it therapeutic concentration, had to be higher than radiological concentration because the treatment energies are higher than the energies used in diagnostics to achieve the therapeutic dose inside the patient. As commented, higher energies mean a more likely Compton scattering contribution against photoelectric events, which reduces the cross-section or interaction probability between the incident beam and the AuNP.

It is clear that before clinical implementation, it will be necessary to consider a toxicity threshold in nontarget organs such as the liver and bladder, limiting the potential concentration values to be administered. These observations are only possible through an in vivo experimental methodology based on animal models. In this work, we consider the potential influence of the enhanced dose on the organs at risk (OARs) involved in actual clinical cases by means of the treatment planning optimization based on the images that include the proposed AuNP concentration modelized as a new material with Monte Carlo simulation method implemented in the in silico stage. In this stage, key parameters involved in the problem were evaluated to find a therapeutic beam with adequate energy to be optimal for dose enhancement. Finally, an in vitro stage was also performed to evaluate the potential radiosensitization of tumor cells with the proposed AuNP concentration under our recommended beam energy.

All these considerations led us to propose a sequential integrated strategy that ethically keeps the idea of the three Rs rule for experimental animal studies—replace, reduce, and refine.

### 2.1. Radiological Concentration Determination

Commercial gold nanoparticles from Nanoprobes (Yaphank, NY, USA, ref: 1115A-5 × 40MGAu) were acquired. These were spheroidal nanoparticles with a core diameter of 15 nm stabilized with a highly soluble and biocompatible organic shell. Different seriated dilutions of gold nanoparticles were prepared in 0.2 mL plastic tubes and their contrast levels were tested in the context of a clinical CT scan for oncology patients. The scanner was a SOMATOM Confidence^®^ of Siemens Healthineers. AuNP were placed in a CIRS 062 electron density calibration phantom, which has other inserts that simulate the physical densities of the following tissues: lung (inhaled), lung (exhaled), adipose, breast (50% adipose/50% glandular), muscle, liver, trabecular bone (200 mg·cm^−3^) and dense bone (800 mg·cm^−3^). The detailed composition for each material is detailed in Appendix A. Its structure also simulates water density, and therefore, when properly placed, and with the disposition observed in Figure 1a, the contrast level that AuNP would provide in a clinical context can be observed, i.e., the contrast level that AuNP would give within a standard patient if the proposed concentrations were achieved. For proper alignment and placement due to the reduced sample volume, the gold nanoparticle probes were settled inside a 50 mL Falcon container, previously filled with a ‘Transonic Gel’ ultrasound gel from TELIC SAU, which presents a water-like electron density and helped fix the positions of the probes inside the Falcon tube, as seen in Figure 1f. Different energies of the X-ray spectra were evaluated.

To determine the concentration value of AuNP that also allows an IGRT treatment, images of the phantom with AuNP were acquired with the Mega Voltage Cone Beam CT (MV-CBCT) photon beam installed on a Siemens ONCOR™ accelerator. The images were acquired with the phantom’s ‘body’ and ‘head and neck’ configurations (Figure 1b,c).

The minimum radiological concentration of AuNP that allows the observation of a contrast of the nanoparticles with respect to soft tissue in both image modalities was determined. Then a larger sample—approximately 2 mL—was placed in a Falcon tube and used to perform a CT and MV-CBCT calibration and to observe the contrast level obtained with the proposed AuNP concentration (Figure 1d,e).

The data obtained were later analyzed using CARMEN software, a program developed under MATLAB from the MathWorks^®^ programming environment. In this software, image processing tools, as well as analytic tools, are implemented for treatment planning, verification, and evaluation [34,35,36].

### 2.2. Evaluation of AuNP Dose Enhancement—In Silico Stage

Simulations of the physical problem were carried out using the Monte Carlo method. This is the gold standard procedure for dose calculation in RT, as it has excellent experimental support at the recurrent energies used for diagnosis and therapeutic purposes. Nowadays, more efficient computational resources permit sampling of event probabilities large enough for reducing the statistical error below 1%. The Monte Carlo code used for this work was EGSnrc [37] for both stages, the particle transport from the linear accelerator to the patient using the BEAMnrc user code [38] and the dose calculation in the patient with BEAMDOSE [39], a modification of DOSXYZnrc [40] developed by our research group to know the contribution of each beam to a single voxel, a necessary step for treatment planning approaches. All these tools are implemented in the CARMEN platform to enable the possibility of RT treatment planning in cases in which the proposed AuNP concentration was virtually included in the therapeutic target volumes defined by the physicians in the corresponding image data.

#### 2.2.1. Linear Accelerator Modeling and Beam Characterization

As stated in previous sections, the beam modality is one of the critical parameters involved in this study. Therefore, an explicit geometrical beam modeling was performed utilizing Monte Carlo simulation.

Two different beams from Siemens ONCOR™ accelerator were used: the clinical 6 MeV photon beam and the MV-CBCT photon beam routinely used for IGRT. While the first beam is currently used to treat patients, the MV-CBCT beam is only used for image acquisition purposes in the treatment room to correct positional changes in the patient. Although this beam is not commercially optimized for therapy use, its lower nominal energy—4.2 MeV, producing an average energy of around 1.4 MeV at the isocenter level, made it interesting to study whether these energies were more efficient in combination with nanoparticles due to their higher photoelectric contribution than the usual therapeutic beams with higher energy. This beam, unlike keV photon beams often studied with rodents, could also be used for treatment as it has relatively high energy that should be able to yield higher radiosensitization than conventional clinical beams [31]. Moreover, in order to check our methodology, this beam could prove the dependence of dose enhancement with beam modality by comparing it with the 6 MeV beam.

All radiation-interacting components of the linear accelerator were modeled employing the Monte Carlo EGSnrc code [37], with the BEAMnrc [38] user code, using the technical specifications of the linear accelerator provided by the manufacturer under a non-disclosure agreement. Then, using MC simulations, the explicit transport of the beams along the linear accelerator head was tested, and the results were compared with experimental dosimetric measurements taken with ionization chambers in the accelerator bunker. Measurements taken in water phantoms usually include a percent depth dose, and dose profiles on the X- and Y-axes. Once these experimental measurements matched the theoretical physical simulation obtained in our laboratory as the result of the MC simulation in a water-based phantom as the target, the correct beam modeling was confirmed.

Siemens ONCOR™ 6 MeV clinical beam has been already modeled by our research group for previous works [2]. MV-CBCT beam modeling implied relevant geometrical changes in the linear accelerator head. Following the specifications provided by the manufacturer, the target on the accelerated electrons collider was replaced by a different target made up of carbon and the flattening filter was removed.

Monte Carlo simulations were performed to generate a percent depth dose (PDD) register. An experimental PDD was taken by using a standard ionization chamber placed in a water phantom following conventional protocols for periodic checks of beam energies. The curve corresponding to these experimental measurements was plotted and compared with the relative dose distribution calculated using our MC-based approaches to check the accelerator modeling and beam characterization of this MV-CBCT photon beam used for IGRT.

#### 2.2.2. AuNP Medium Modeling and Dose Enhancement Simulations

Once the radiological concentration value was determined, the next step was to find out whether the same concentration could also be useful for therapeutic purposes. Otherwise, this initial concentration would be multiplied by an integer number to find a significant dose enhancement by simulating with Monte Carlo as it is explained below.

Using this methodology, we performed several simulations of clinical cases detailed later. We used the patient image data involved in the actual treatment planning of the clinical cases by virtually adding nanoparticles to the BTV region, previously defined using the SUV present in the PET study, using a semi-automatic segmentation algorithm [41]. Therefore, the radiological concentration was located in the planning image in voxels labeled as part of the tumor, supposing a biodistribution efficiency of AuNP similar to the PET capacity to identify the extension of the disease, which is not really expected, but it was considered as a starting point to follow the methodology stated above.

To consider the AuNP in the calculation of the MC dose, a new material was simulated. Considering that the tumor can be radiologically close to soft tissue, the AuNP material was generated as the result of adding the appropriate amount of gold to the soft tissue material with a similar physical density. At this point, three options for creating material can be used: element, when there is only one type of atom implied in the material; compound, when there are different types of atoms and the proportions of each atom are given with the relative number of atoms; and mixture, similar to compound but when the proportions are given by mass. On the basis of the above reasoning, the mixture was the option applied for the new material corresponding to the tumor containing AuNP. The compositions of the materials used can be accessed in Appendix A.

The simulation parameters were chosen to explicitly carry out the transport of particles along the densities and materials included in the patient image to achieve precise calculations, which is accompanied by a higher computational cost. Fortunately, our laboratory has parallel computing resources, distributing complex tasks and, therefore, reducing elapsed time.

Once the simulation parameters were set up, we hereby used a clinical case of head and neck cancer with morphological information from CT imaging, functional information from PET studies, and also information about the target regions defined using a semi-automatic segmentation algorithm [41] and OARs structures using routine clinical protocols. These OARs, attending to the tumoral localization, were submaxillary muscles, parotids, thyroid, larynx, and spinal cord.

Doses delivered in the BTV with and without nanoparticles were compared. These changes were evaluated in the form of isodoses and dose distribution using isolines, dose-volume histograms, dose enhancement factor (DEF), and conformity number (CN), a relative measure of dosimetric target coverage and the preservation of normal tissues in a treatment plan [42].

DEF and CN were calculated following Equations (1) and (2), respectively:(1)$$\mathrm{DEF}=\frac{\mathrm{Dose}\;\mathrm{in}\;\mathrm{target}\;\mathrm{with}\;\mathrm{AuNP}}{\mathrm{Dose}\;\mathrm{in}\;\mathrm{target}\;\mathrm{without}\;\mathrm{AuNP}}$$
(2)$${\mathrm{CN}}_{\%}=\frac{\;\frac{\mathrm{BTV}\;\mathrm{voxels}\;\mathrm{receiving}\;\%\;\mathrm{of}\;\mathrm{prescription}\;\mathrm{dose}}{\mathrm{BTV}\;\mathrm{voxels}}\;}{\;\frac{\mathrm{BTV}\;\mathrm{voxels}\;\mathrm{receiving}\;\%\;\mathrm{of}\;\mathrm{prescription}\;\mathrm{dose}}{\;\mathrm{Voxels}\;\mathrm{receiving}\;\%\;\mathrm{of}\;\mathrm{prescription}\;\mathrm{dose}\;}\;}$$

RT treatment planning was carried out with the therapeutic 6 MeV beam. This was the energy used for the original dose painting treatment planned for the patient. Additionally, treatment planning was optimized using the tools implemented on the CARMEN platform, to observe possible improvements in treatment due to the presence of AuNP.

In addition to the simulations described with the 6 MeV beam, several treatment plannings were carried out with the MV-CBCT beam. The simulations involved different conditions to evaluate the value established in the radiological AuNP concentration determination.

### 2.3. Evaluation of AuNP Radiosensitization—In Vitro Stage

As discussed in the introduction, determining the dose enhancement factor is not enough to completely predict the radiobiological effect of the use of AuNP in combination with RT. Once the beam spectra that provides interesting DEF values were found, they were used to evaluate DNA damage in human cancer cells irradiated with different doses, with and without nanoparticles. DNA damage in cells was labeled through γH2AX foci immunofluorescence and observed using super-resolution microscopy. The acquired images were quantified using automatic computational procedures in MATLAB R2022a (Natick, MA, USA) environment.

#### 2.3.1. Cell Culture and Irradiation

Human cancerous cells MDA-MB-231 were used in this study. Cells were maintained under the usual required cell culture conditions, grown in Dulbecco’s Modified Eagle Medium (DMEM) with high glucose and pyruvate [ThermoFisher, Paisley, UK, ref: 21969035] supplemented with 4 mM L-glutamine [ThermoFisher, Paisley, UK, ref: 25030081], 10% fetal bovine serum (FBS) [Sigma-Aldrich, Darmstadt, Germany, Ref: F7524-100ML], and penicillin (100 U·mL^−1^), streptomycin (100 µg·mL^−1^), and amphotericin B Gibco (0.25 µg·mL^−1^) [ThermoFisher, Grand Island, NY, USA, ref: 15240062].

The cells were used for the experiments in their fourth passage. They were grown in 12 mm circular coverslips treated with poly-L-lysine [Sigma-Aldrich, Darmstadt, Germany, ref: P1524-25MG], and placed on the bottom of 24-well cell culture plates. The four central wells in five plates were used to grow the cells, totaling 20 wells. In each plate, two wells contained complete DMEM, while the other two wells contained complete DMEM and 15 nm Aurovist AuNP at ~22.3 mg Au·mL^−1^ media, our proposed AuNP concentration obtained in the previous stage [Nanoprobes, Yaphank, NY, USA, ref: 1115A-5 × 40MGAu]. After the nanoparticles were added, an incubation of 12 h took place.

After incubation, cells were placed on the bunker treatment bed and irradiated using Siemens Oncor MV-CBCT IGRT beam. Each cell plate constituted a different irradiation dose condition for the study. The plates were irradiated from the bottom to avoid air gaps in the incidence path, and placed on top of 1 mm of solid water phantom, at the maximum of the percentage depth dose for that beam spectrum. Solid water was also placed on top of the cell plate lids, to reduce the impact of a possible build-down effect. Cells were irradiated with 0.25 Gy, 0.5 Gy, 1 Gy, and 2 Gy. A plate remained unirradiated, as a control plate.

#### 2.3.2. Immunofluorescence Protocol

Thirty minutes after irradiation, the cells were fixated using a 4% paraformaldehyde [Sigma-Aldrich, Darmstadt, Germany, ref: P6148-500G] solution in PBS [Sigma-Aldrich, Darmstadt, Germany, ref: P4417-50TAB] for 10 min. They were washed three times for 5 min in agitation and then, permeabilized using a 0.1% Triton-X 100 solution [Fisher Bioreagents, Madrid, Spain, ref: BP151-500] for 10 min. Next, a blocking solution consisting of Triton X-100 0.05% and BSA 3% [Sigma-Aldrich, Darmstadt, Germany, ref: A9647-50G] was applied for 30 min.

After this preparation, a primary antibody targeting the phosphorylated Ser-139 of H2AX [Novus, Madrid, Spain, ref: NB100-74435] was added at a concentration of 2 µg·mL^−1^ in a PBS solution along with Triton X-100 0.05% and BSA 3%. Following 2 h of incubation at room temperature, the coverslips were washed three times for 5 min in agitation using a PBS solution with Triton X-100 0.05% + BSA 3%.

Subsequently, incubation with the secondary antibody Alexa Fluor 488 [Invitrogen, Eugene, OR, USA, ref: A-21121] took place, at a 1:1000 concentration in a PBS solution with Triton X-100 0.05% and BSA 3%. After one hour of incubation at room temperature, the coverslips were washed three times with PBS for five minutes, and the coverslips were mounted with Fluoromount + DAPI [Invitrogen, Carlsbad, CA, USA, ref: 00-4959-52].

#### 2.3.3. Image Acquisition

Each of the preparations was scanned with the Leica Stellaris 8 confocal microscope using the stimulated emission depletion (STED) super-resolution technique. All images were taken with the HC PL APO CS2 100×/1.40 OIL objective.

A sequential scan was made, saving information in two channels. The blue channel was assigned to the DAPI signal corresponding to the cell nucleus, with the pinhole set at 1 AU, excitation at 405 nm with intensity ~10%, and a detector HyD S 1 (430–499 nm) with the gain set to 24. The green channel recorded the information of the γH2AX foci, with the pinhole set at 1 AU, excitation at 499 nm with intensity ~6.7%, and a detector HyD S 3 (504–581 nm) with the gain set to 16. For the foci channel, STED was used, with a 592 nm depletion laser with intensity set to 20%.

A wide tile scan composed of at least 16 single image fields was taken for each sample, obtaining 2 tile scan images per irradiation condition, both for control and nanoparticle-treated cells. Each experimental condition consisted of at least 200 cellular objects and up to 360 objects.

The pixel resolution of the single image fields was optimized according to Nyquist’s law. The resolution for each image constituting the tile scan was 1880 × 1880 px, and the pixel depth was set to 8 bits for each channel. The signal detected in the γH2AX channel was averaged from 3 different scans, to reduce signal noise.

#### 2.3.4. Image Processing and Quantification

For this study, we developed a robust automatic quantification software, which we called SoFoCo, an adaptation of the FoCo algorithm by Lapytsko et al. [43]. Unlike FoCo, our SoFoCo software [v1.0, Seville, Spain] was coded entirely in MATLAB, and the DNA damage quantification levels are not measured in terms of the absolute number of foci per cell nucleus, which is an operation highly dependent on the nucleus size, the height at which the Z-slice was taken in the confocal microscopy, and on cell labeling algorithm used to separate cell groups into individual cells. Rather, we quantified the foci density per object, whether they are individual cells or small groups not separated by thresholding. This way, some steps are modified or suppressed, such as the usage of watershed for labeling individual cells, or the need to eliminate objects at the borders of the image. An illustrative workflow of the followed process is available in Figure 2.

In short, the process consists of the separation of the foci and nuclei channel. In the nuclei channel, a threshold is applied to binarize the image, in our case with the Otsu method [44]. Then, an adaptive median filter is applied to denoise the image, holes in the objects are filled, and a morphological opening by reconstruction is performed to delete small objects that are not nuclei, considering a minimal nuclei radius of 25 pixels for our image dataset. After that, a morphological closing is applied by dilating, filling holes, and eroding. The dilations and erosions were performed 4 times. The objects in that last mask are used as the individual areas in which the foci density will be counted. For that, the object area in the foci channel is filtered with an adaptive median filter and the top-hat transform is applied with a maximal focus radius *r_f_* for reducing background local noise. Then, the H-maxima transform is used to find local maxima with the threshold level of Otsu in the foci channel. All those regional maxima are counted as foci if their intensity value in the original image is higher than the user-defined foci intensity threshold *T_e_*. *r_f_* value used was 23 and 13 for control and AuNP datasets, respectively. *T_e_* value was 0.55 for both datasets.

The values of the user-defined parameters *r_f_* and *T_e_* were determined through mathematical optimization as stated by Ivashkevich et al. and Lapytsko et al. [43,45]. The tested values for *r_f_* and *T_e_* optimization were {9, 11, 13, 15, 17, 19, 21, 23, 25, 27} and {0.3, 0.35, 0.40, 0.45, 0.5, 0.55, 0.6, 0.65, 0.7}, respectively. The optimization procedure was performed individually for the AuNP-treated cells and the control cells, as the sole incubation with nanoparticles constituted a different culture condition that could modulate the foci. Optimization results are detailed and can be accessed in Appendix A.

The quantification algorithm and code used can be accessed through the repository https://github.com/josealv23/SoFoCo. We intend to update the program with more complex features in the future, considering also user feedback.

#### 2.3.5. Statistical Analysis

DNA damage density was calculated in terms of the number of foci in an object of quantification divided by the area, in pixels, of that object. The average foci density of each object was calculated for each dataset. For every image, outliers were detected and excluded for further analysis. Outliers were defined as the objects with foci densities ranging out of [Q1—1.5·(Q3–Q1), Q3 + 1.5·(Q3–Q1)]. After outliers were excluded, the average foci density was calculated for each condition, along with the standard error of the mean.

The data was plotted against the doses for each dataset, control and AuNP, and a linear trend was calculated. The correlation value was calculated and statistically tested with the Pearson’s R test, having previously run the Shapiro–Wilk normality test on the data.

## 3. Results & Discussion

### 3.1. Beam Modeling

The beam was properly modeled according to the provided technical specifications. The percent depth dose (PDD) generated using MC simulations was plotted and compared with the experimental PDD, and this comparison achieved high agreement, as reported in Appendix A.

This result gave confidence for the following MC simulations in this work. The agreement obtained was essential since any small mismatch would result in some changes in the beam spectrum, therefore affecting the beam cross-section due to the presence of AuNP.

### 3.2. AuNP as Diagnostic Agents

The CT and MV-CBCT images of the electron density phantom with a dilution of AuNP in the center slot were acquired. The CT data obtained suggested that energies of the X-ray spectra such as 100 or 120 keV are preferable to 80 keV, which was expected because 80 keV is slightly lower than the gold K-edge, whereas 100 and 120 keV beam particles do have enough energy when reaching the gold, in the center of the phantom. The CT calibration curve is detailed in Appendix A.

An AuNP concentration value of 2.23 mg Au·mL^−1^ solution allowed to obtain CT images with a radiological contrast level sufficient to distinguish between the soft tissue and the AuNP dilution, in the phantom with body configuration, as shown in Appendix A. However, it was necessary to increase this concentration 10 times to 22.3 mg Au·mL^−1^ solution, and also to use the head and neck phantom configuration to obtain an adequate contrast level in the MV-CBCT images. Moreover, this higher concentration was also necessary to obtain a significant dose enhancement for therapeutic purposes, as it will be discussed later.

The CT and MV-CBCT images with an AuNP concentration value of 22.3 mg Au·mL^−1^ solution are shown in Figure 3. In the CT images, the pixel intensity value of the AuNP dilution was approximately 740 Hounsfield Units (HU), while the values of the trabecular and dense bone are 230 HU and 910 HU, respectively. Therefore, the location of AuNP with this concentration would be easily detected (Figure 3a) without the need to apply any window to the images. However, as part of the clinical routine, soft tissue and bone radiological window were applied to highlight the contrast level of the AuNP. Figure 3d shows the MV-CBCT image of the same phantom configuration as Figure 3a. In this case, even by applying different windows (Figure 3e,f), it was not possible to detect the location of AuNP with the same concentration. The AuNP dilution was located in the center of the phantom, so the signal received by the detector was too weak due to the thickness of the material through which the particles of the beam with that specific energy must pass in the body configuration phantom. Figure 3g shows the MV-CBCT image of the head and neck configuration of the phantom. Here, the desired contrast was differentiable. The pixel intensity value of the AuNP dilution was 187 equivalent HU, between the values of trabecular and dense bone, 153 and 348 equivalent HU, respectively. The radiological contrast in these conditions was enough to detect AuNP and therefore showed to be an efficient tool for monitoring disease in the radiation bunker after previous delivery treatment. Figure 3h,i show the soft tissue and bone radiological window of the same image.

However, even though AuNP has a high value as a contrast agent at the concentration evaluated, this contrast capacity remains less powerful than previous results in the scientific literature. The main reason for this disagreement is that clinical conditions were applied in this work, whereas in most of the published literature, AuNP contrast is measured using nano-CT, or acquisitions lack the positioning of AuNP in a clinical context, i.e., AuNP is not surrounded by the tissue-equivalent medium. Therefore, the contrast power of AuNP is ensured, but with the concentration required is higher than the one proposed in previous works. This new AuNP concentration could compromise its pharmacokinetic and toxicological possibilities unlike other lower concentrations used before in rodents within nano-CT, but the size differences between these animals and humans allow us to be optimistic at this point. Previous studies on the toxicological effects of these AuNP have not shown toxicity problems with these concentrations. Hainfeld et al. used injections of up to 40 mg Au at concentrations of 200 mg·mL^−1^ per mouse [46]. We expect that the proposed concentration values can be achieved locally in tumor cells considering the low toxicity of the product, with an LD_50_ > 5 g Au·kg^−1^, according to the nanoparticles’ data sheet. Anyway, the possibilities to functionalize these AuNP, augmenting their targeting efficiency, encourage further efforts to transfer our results to the clinic.

These AuNP have long blood circulation times with a stated half-life of ~15 h. This would allow time for treatment planning after image acquisition, and even follow-up images could be acquired after treatment. Nebuloni et al. stated that an anatomical representation of blood vessels could be possible at 2 and 3 days after injection in mice. This generates expectations of using MV-CBCT for follow-up imaging without additional injections [47].

### 3.3. AuNP as Therapeutic Agents

Differences found in planning with and without nanoparticles for the 6 MeV photon beam—the original treatment planning of the actual cases—were almost negligible. The mean DEF for the concentration evaluated is shown in Table 1, and no significant dose enhancement was found for this beam energy. As dose enhancement measures absorbed radiation, a physical process, substantial differences are not expected for other cases of similar characteristics using the same beam energies and AuNP concentrations. As stated in the Introduction section, despite these small changes in dose enhancement, the use of 6 MeV beams could be worthwhile from the point of view of improvements in radiosensitization since low-energy events involved such as Auger electron emission and others that promote a higher number of ionization events, increasing oxidative stress caused by newly generated ROS. In any case, we ended up not carrying out the radiobiological effect experiment for this energy beam and focusing our proposal on a lower energy beam such as the one provided by MV-CBCT.

The target dose definition was parametrized using the previously described CN, showing CN_100_ scores of 0.4145 and 0.3889, while CN_90_ scored 0.2442 and 0.2403, with and without nanoparticles, respectively, for both indexes (see Table 1). Therefore, the target dose conformation was improved by the presence of AuNP, even at beam energies of 6 MeV. This agrees with the small dose augmentation observed. In any case, these changes in dose distribution are clearly not enough to have clinical relevance, at least in terms of absorbed dose, and therefore optimizing the treatment would not provide the desirable dose reductions to OAR.

In contrast with the results obtained with a 6 MeV photon beam, the experiments carried out with the MV-CBCT beam yielded some significant results. The results of the 22.3 mg Au·g^−1^ tissue concentration are also shown in Table 1. First, the average DEF was 1.159, with a standard deviation of 0.021. This is of great interest, as it gives an increase in absorbed dose of ~16%. This dose enhancement affects mainly the target, although some very close OARs also showed that their dose slightly increased. The isodoses of the planning with AuNP showed a high conformation of the target with the prescribed dose meanwhile avoiding OARs, as can be seen in Figure 4. The same treatment planning was simulated without AuNP, shown in Figure 4. As seen, the planning with AuNP achieves an excellent result that could not be possible without the presence of nanoparticles.

Treatment planning without AuNP was optimized to try to reach the dose objectives and was again compared with the optimized planning in presence of AuNP (Figure 5). It can be observed that with the same optimization objectives, these were achieved when AuNP were present, whereas the prescription dose was scarcely delivered to the BTV when AuNP were absent. The dose conformation, facilitated by AuNP’s local dose enhancement, avoided the generation of ‘hot spots’ and delivered high doses with precision to target volumes. The CN factor was parameterized, resulting in CN100 values of 0.6525 with nanoparticles versus 0.3422 without them, which would give highly significant clinical differences (see Table 1).

### 3.4. AuNP Radiosensitization

Images of the γH2AX foci immunofluorescence were acquired successfully using the STED super-resolution technique. Some representative fragments of the tile scans obtained are shown in Figure 6. A trend of higher foci density can be observed when the doses increase, as expected. Despite a higher γH2AX foci density can be slightly observed when nanoparticles are present, especially at higher doses, the differences between control cells and AuNP-treated cells are not clear without quantifying the images. The complete image dataset is available in the Appendix A.

Quantification using the SoFoCo algorithm confirmed the relationship between higher γH2AX foci density and higher radiation doses, as can be seen in Figure 7. Linear regression for both datasets yielded R^2^ values 0.9847 and 0.9977 for the control and AuNP data, respectively. Both correlations were statistically significant with a *p*-value < 0.001.

Given that the last incubation with DMEM or AuNP, for control and AuNP datasets, respectively, changes the culture conditions, the data in Figure 7a,b are not directly comparable. However, analyzing the increase in relative foci density between each group using the gradients of the linear regression equation, it can be seen that the DNA damage density increases more significantly with nanoparticles, with a gradient of ~3.25 (see Figure 7b), than without nanoparticles, with a gradient of ~2.23 (see Figure 7a).

Thus, it could be deduced that nanoparticles at a concentration of ~22.3 mg Au·mL^−1^ increase the radiosensitization effect by a factor of ~1.46, resulting from the comparison of both gradients. This factor is considerably higher than the calculated dose enhancement factor for the clinical case. Hence, the implementation of AuNP under these conditions could represent a clear radiobiological advantage beyond dose distribution benefits.

## 4. Conclusions

According to the results obtained, the established AuNP concentration value of around 20 mg Au·mL^−1^ could be used both for diagnosis and for its application in IGRT treatments, under the conditions of the clinical routine. To the extent that AuNP are conveniently retained by the tumor mass, IGRT would provide a functional image beyond the usual one to correct patient positioning, which would mean monitoring the course of the disease throughout the radiotherapy treatment sessions, at least, for cases with tumors that are not too deep, such as the typical ones in the head and neck and others, such as in breast.

Regarding the toxicological aspects, as discussed, this concentration is lower than those previously used for animals. In humans, the injection volume would be different. Hence, the infused concentration could be smaller than our proposed concentration in the target, as long as the biodistribution of AuNP is efficient enough.

Anyway, the dose enhancement showed could be implemented in treatment planning for dose distribution calculation and increasing the dose in the PTV while decreasing the dose in OAR. Furthermore, this dose enhancement is delivered at the cell level, so AuNP implementation is part of the role of the precision medicine in radiotherapy.

The proposed AuNP concentration has shown therapeutic efficacy as radiotherapy dose enhancer with an energy beam close to 1 MeV, such as the MV-CBCT beam for IGRT. In these conditions, dose enhancement was increased to ~16% based on in silico experiments carried out with accurate modeling of the physical problem. In vitro experiments based on γH2AX foci methodology showed even more exciting results on radiosensitization by achieving a 46% improvement.

This result is relevant insofar as it was achieved under clinical conditions and beam energy configuration that could even provide a treatment monitoring procedure. These results now encourage an in vivo evaluation on a physiologically and dimensionally convenient animal model, having notably reduced the number of sacrifices that would have been necessary to achieve the results of this work without following a sequential integrated strategy.

## Figures and Tables

**Figure 1 biomedicines-10-01214-f001:**
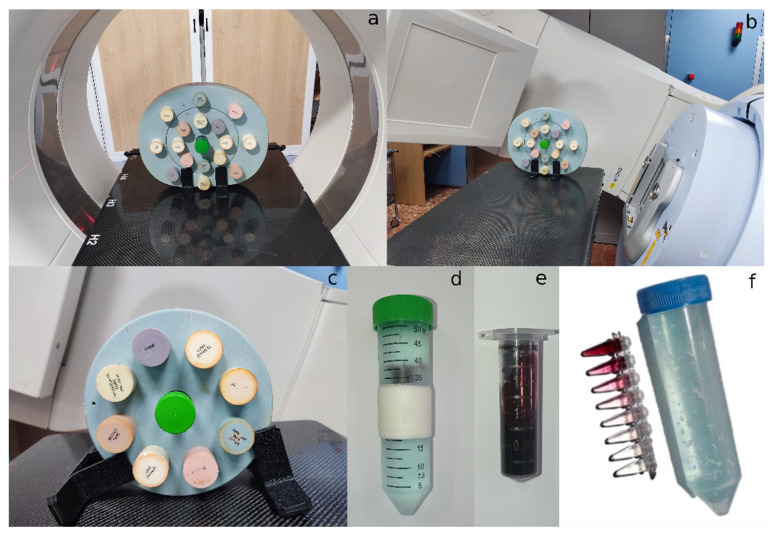
Experimental setup for the determination of radiological concentration: (**a**) general view of the CIRS 062 electron density phantom (body configuration) with AuNP placed in the center slot, inside a CT scanner; (**b**) general view of the same phantom with AuNP on a radiotherapy couch, for MV-CBCT image acquisition; (**c**) head and neck configuration of the phantom with AuNP; (**d**) falcon tube filled with ultrasound gel containing the nanoparticles cryovial with a concentration of 20 mg Au·mL^−1^ solution, shown in (**e**) with a closer zoom; and (**f**) Seriated concentrations and the Falcon tube filled with ultrasound gel.

**Figure 2 biomedicines-10-01214-f002:**
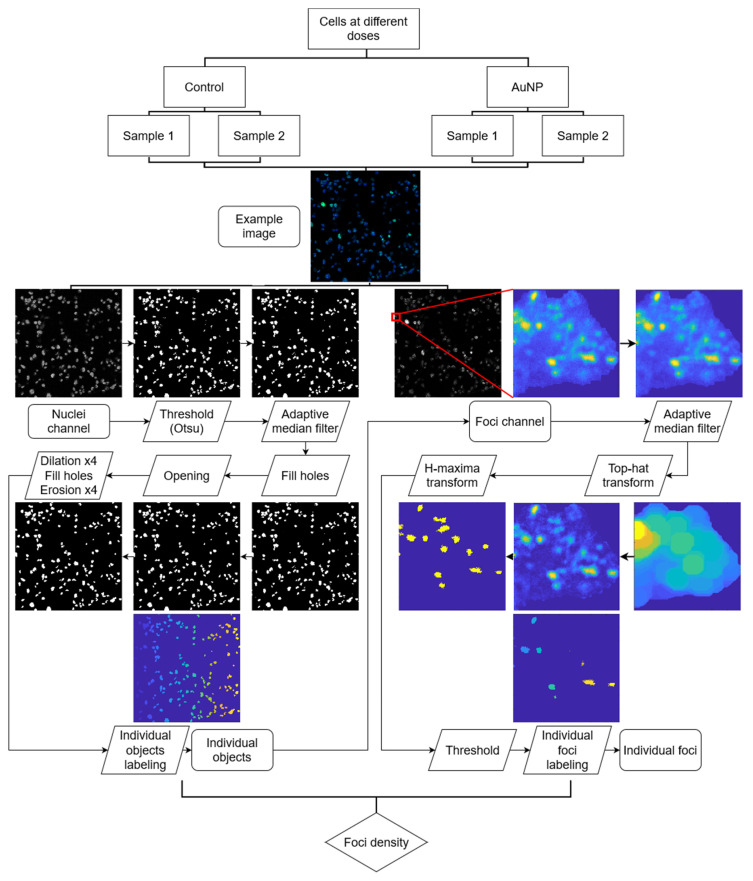
Workflow of the SoFoCo algorithm used for foci density quantification.

**Figure 3 biomedicines-10-01214-f003:**
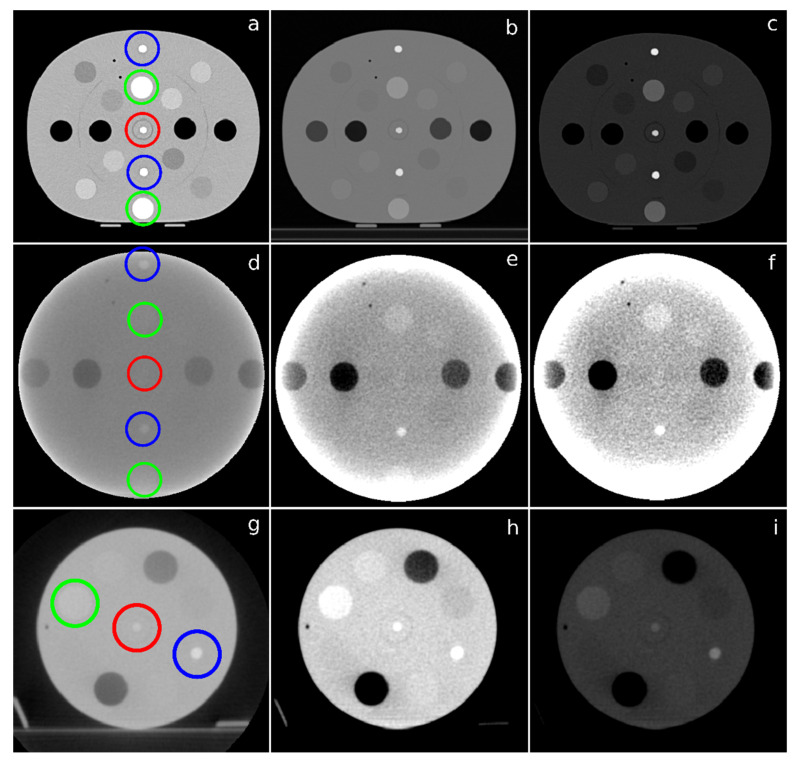
(**a**) axial CT slice of the phantom with body configuration and 22.3 mg Au·mL^−1^ solution concentration of AuNP in the center slot, and the same slice after applying a soft tissue; (**b**) and a bone; (**c**) radiological window. The X-ray spectra energy was 120 keV. (**d**) MV-CBCT image of the phantom with body configuration and 22.3 mg Au·mL^−1^ solution concentration of AuNP in the center slot, and the same image after applying a soft tissue; (**e**) and a bone; (**f**) radiological window; (**g**) MV-CBCT image of the phantom with head and neck configuration and 22.3 mg Au·mL^−1^ solution concentration of AuNP in the center slot, and the same image after applying a soft tissue; (**h**) and a bone; and (**i**) radiological window. Red, green, and blue circles indicate the location of the Falcon tube with AuNP, trabecular bone insert, and dense bone insert, respectively.

**Figure 4 biomedicines-10-01214-f004:**
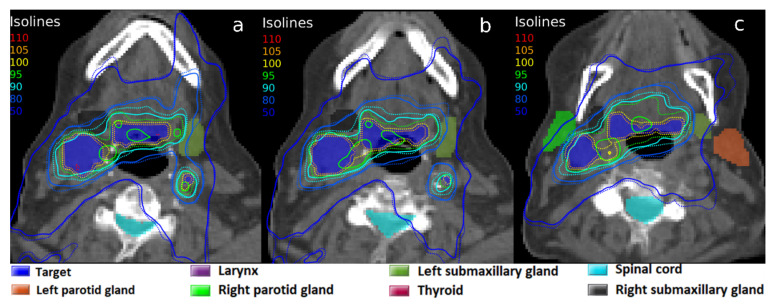
Isodoses in three axial CT slices (**a**–**c**) corresponding to the same radiotherapy treatment planning with the MV-CBCT beam: dose calculation without AuNP (solid line), and with a simulated 22.3 mg Au·g^−1^ tissue concentration of AuNP in the target volume (dotted line). 100% isodoses represent the clinical prescribed dose of 70 Gy.

**Figure 5 biomedicines-10-01214-f005:**
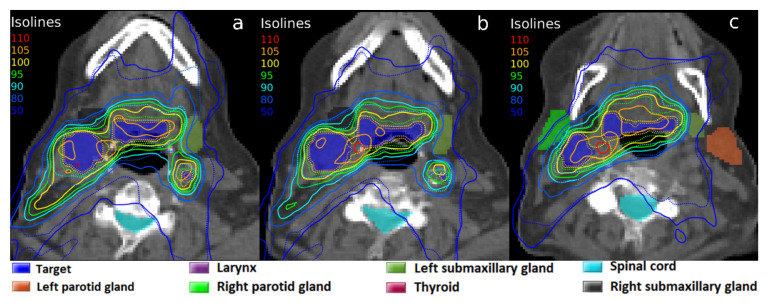
Isodoses in three axial CT slices (**a**–**c**) corresponding to the radiotherapy treatment planning with the MV-CBCT beam, optimized for each configuration: without AuNP (solid line), and with a simulated 22.3 mg Au·g^−1^ tissue concentration of AuNP in the target volume (dotted line). 100% isodoses represent the prescribed dose of 70 Gy.

**Figure 6 biomedicines-10-01214-f006:**
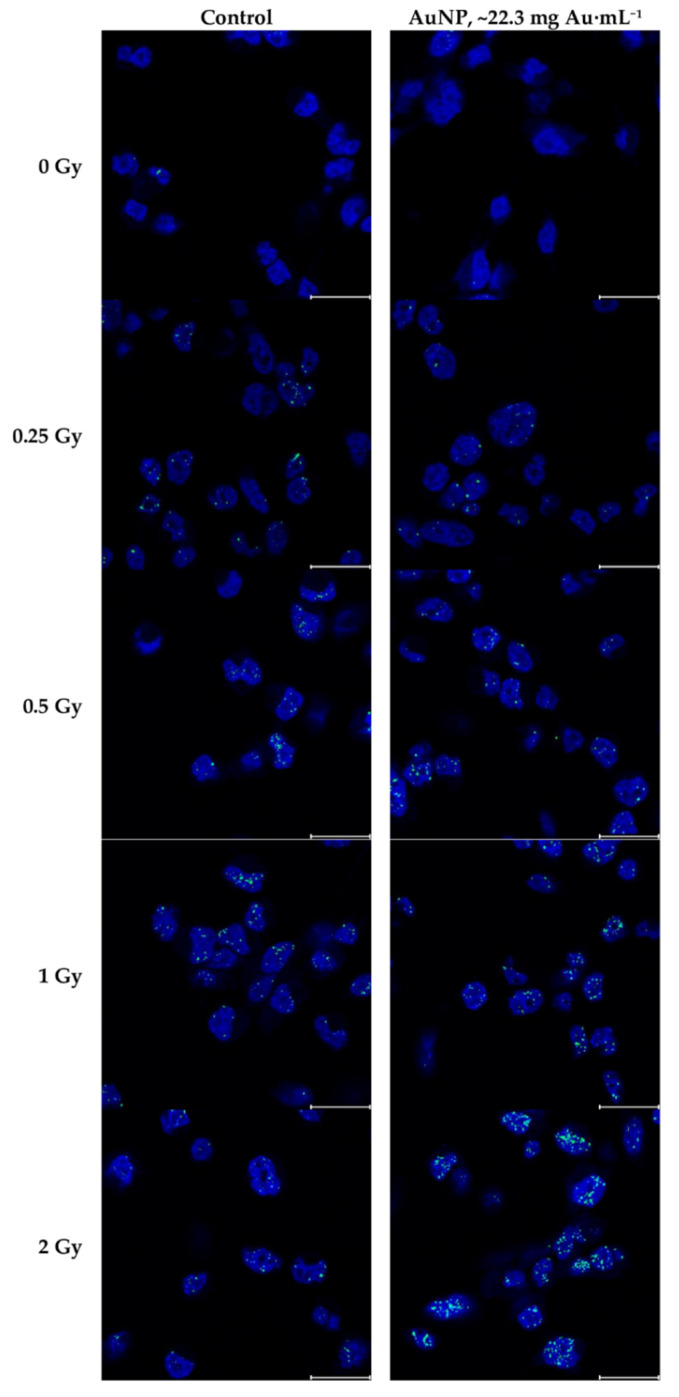
Fragments of the immunofluorescence images analyzed, both control and AuNP-treated, irradiated under different doses. Blue channel represents nuclei with DAPI staining, green channel represents γH2AX labeled with Alexa Fluor 488 secondary antibodies. Blue channel window level is [0 255]; green channel window level is [140 255], representing the window level applied for foci quantification. The scale bar corresponds to 20 μm.

**Figure 7 biomedicines-10-01214-f007:**
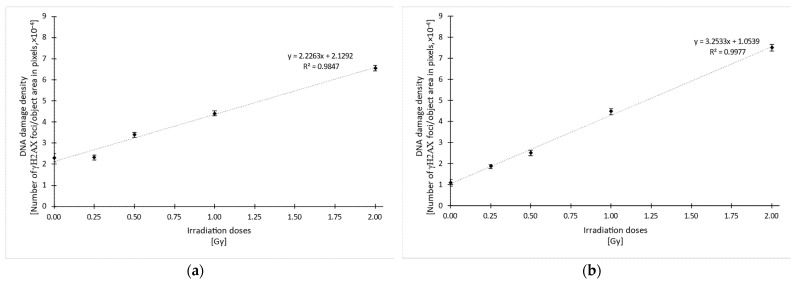
Quantified DNA damage density in terms of the number of γH2AX foci per object area, in pixels, for each irradiation dose condition, for control cells (**a**) and AuNP-treated cells (**b**). Error bars represent the mean standard error. The linear regression equation is shown in the graph, along with the R^2^ value. R^2^ is statistically significant, with *p* < 0.001.

**Table 1 biomedicines-10-01214-t001:** Data summary of evaluation parameters for each simulation.

Energy Beam		D_max_ (Gy)	CN_100_ after Optimization	CN_90_ after Optimization	DEF (μ ± σ)
6 MeV	without AuNP	80.5	0.3889	0.2403	1.009 ± 0.015
	with AuNP	81.0	0.4145	0.2442	
MV-CBCT	without AuNP	70.9	0.3422	0.2133	1.159 ± 0.021
	with AuNP	78.5	0.6525	0.2747	

## Data Availability

The data presented in this study are available within the article and its Appendix A. Further data are available on request from the corresponding author.

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
