# Peer review of "Clinical Feasibility Study of Gold Nanoparticles as Theragnostic Agents for Precision Radiotherapy"

_biomedicines, 2022, doi:10.3390/biomedicines10051214_

Round 1

Reviewer 1 Report

In this work the Authors proposed a very interesting study about the use of gold nanoparticles as theragnostic agents. The topic is certainly of great scientific interest. The manuscript is very well written.

Specifically, the introduction scenario in which this research arises is well illustrated, the reasons driving the research and the aim of the study are totally clear and well provided. The presented introduction and the method used for the proposed study have been totally appreciated.

Just as a minor request, some comments and additional information about the AuNPs should be reported. It is well known that such metal nanoparticles are very useful for this kind of application. However, as the Authors mentioned in the introduction, there are some critical aspects about the AuNP use, which represent a significant limit in the final biomedical applications of AuNPs: size, shape and concentration. Could the Authors add some details about the size and the shape (probably spheroidal) of the AuNPs that they used? Furthermore, is the minimum radiological concentration of AuNP also a safe concentration for the human? What do the Authors think about the toxicological effect of the use of 20 mg Au·mL-1? Moreover, could there be some long-term effects (e.g. AuNP accumulation)? The nanomaterial research community is continuously studying some useful methods to enhance the biocompatibility of metal NPs, for example exploiting the functionalization of MeNPs. However, could the functionalization affect the theragnostic properties? What are the Author opinions about that point?

In conclusion, just minor revisions are necessary for paper publication.

Author Response

Thank you for your comments. We think the manuscript improbed by following your suggestions.

Reviewer 2 Report

The manuscript coauthored by José and colleagues and originating from Dr. Leal’s Lab ostensibly investigated the clinical feasibility of gold nanoparticles as precision radiotherapy drugs through simulation calculations and in vitro experiments. The article explores that the use of AuNP concentration around 20 mg Au·mL-1 was found to be the optimal concentration for imaging and therapy. And compared with 6 MeV photon beam source commonly used in clinic, the low radiation dose photon beam (MV-CBCT) used for image acquisition shows enhanced absorbed dose and more prominent radiosensitization effect of AuNP at the cellular level.

In general, I found the manuscript to interesting, and relatively well-written. I do, however, have some comments and concerns for the authors to address prior to this manuscript becoming acceptable.

  1. "20 mgAu·mL-1" in the abstract lacks spaces. What’s the meaning of "100nm" in "8 hours after injection for 100nm" on line 110? Is the "deexcitations" on line 143 a spelling mistake? There are two periods at the end of the sentence in the annotations of Figures 4 and 5.
  2. The abbreviation that appears for the first time in the article needs to give the full name, such as "TPSs" on line 132, "Rs" on line 215.
  3. The incubation concentration of AuNPs used in Figure 6 should be indicated in the legend.
  4. "An AuNP concentration value of 2.23 mg/ml allowed to obtain CT images … and the AuNP dilution, in the phantom with body configuration. However, it was necessary to increase this concentration 10 times up to 22.3 mg/ml, and … in the MV-CBCT images." This paragraph needs to give more relevant data, such as CT imaging of different materials comparison with 2.23 mg/ml AuNP. How is the multiple of 10 times obtained? Please give the imaging data of multiple concentrations of different multiples.
  5. The use of MV-CBCT as a photon beam can greatly improve the radiosensitization ability of AuNP, but it is mentioned in the article that it must be combined with the phantom of head and neck configuration. If the target is located on the torso or “body”, is there any way to solve it? Is it okay to increase the concentration of Au NPs? Or must a higher-energy photon beam be used?
  6. The article shows that AuNPs concentration around of 20 mg Au·mL-1 can not only be used for image acquisition, but also can obtain a strong radiosensitization effect. In practice, however, reaching such high concentrations at target sites in vivo would be fraught with many difficulties, as would the toxic side effects of high concentrations. Since the aim of the article is to further clinical application, the authors need to provide more biocompatibility data of the synthesized AuNPs, such as cytotoxicity experimental data, etc.

Author Response

(The authors gave the same response as above.)

Reviewer 3 Report

The article entitled Clinical feasibility study of gold nanoparticles as theragnostic agents for precision radiotherapy  is a document of interesting subject matter.

However, it needs some minor changes before being accepted. Make the following corrections:

  1. The manuscript needs to be checked again for journal style.
  2. The abstract and conclusion are a bit too concise. Please make a general abstract and conclusion of the study.
  3. Please explain in the Introduction with more detail about the research novelty.
  4. The objective or objectives should be clearly elucidated in the last paragraph of the introduction.

Author Response

(The authors gave the same response as above.)
